# Regularization Path of Cross-Validation Error Lower Bounds

**Atsushi Shibagaki, Yoshiki Suzuki, Masayuki Karasuyama, and Ichiro Takeuchi**
Nagoya Institute of Technology
Nagoya, 466-8555, Japan
{shibagaki.a.mllab.nit,suzuki.mllab.nit}@gmail.com
{karasuyama,takeuchi.ichiro}@nitech.ac.jp

## Abstract

Careful tuning of a *regularization parameter* is indispensable in many machine learning tasks because it has a significant impact on generalization performances. Nevertheless, current practice of regularization parameter tuning is more of an art than a science, e.g., it is hard to tell how many grid-points would be needed in cross-validation (CV) for obtaining a solution with sufficiently small CV error. In this paper we propose a novel framework for computing a lower bound of the CV errors as a function of the regularization parameter, which we call *regularization path of CV error lower bounds*. The proposed framework can be used for providing a theoretical approximation guarantee on a set of solutions in the sense that how far the CV error of the current best solution could be away from best possible CV error in the entire range of the regularization parameters. Our numerical experiments demonstrate that a theoretically guaranteed choice of a regularization parameter in the above sense is possible with reasonable computational costs.

## 1 Introduction

Many machine learning tasks involve careful tuning of *a regularization parameter* that controls the balance between an empirical loss term and a regularization term. A regularization parameter is usually selected by comparing the cross-validation (CV) errors at several different regularization parameters. Although its choice has a significant impact on the generalization performances, the current practice is still more of an art than a science. For example, in commonly used grid-search, it is hard to tell how many grid points we should search over for obtaining sufficiently small CV error.

In this paper we introduce a novel framework for a class of regularized binary classification problems that can compute *a regularization path of CV error lower bounds*. For an $\varepsilon \in [0, 1]$, we define $\varepsilon$-*approximate regularization parameters* to be a set of regularization parameters such that the CV error of the solution at the regularization parameter is guaranteed to be no greater by $\varepsilon$ than the best possible CV error in the entire range of regularization parameters. Given a set of solutions obtained, for example, by grid-search, the proposed framework allows us to provide a theoretical guarantee of the current best solution by explicitly quantifying its approximation level $\varepsilon$ in the above sense. Furthermore, when a desired approximation level $\varepsilon$ is specified, the proposed framework can be used for efficiently finding one of the $\varepsilon$-approximate regularization parameters.

The proposed framework is built on a novel CV error lower bound represented as a function of the regularization parameter, and this is why we call it as a regularization path of CV error lower bounds. Our CV error lower bound can be computed by only using a finite number of solutions obtained by arbitrary algorithms. It is thus easy to apply our framework to common regularization parameter tuning strategies such as grid-search or *Bayesian optimization*. Furthermore, the proposed framework can be used not only with exact optimal solutions but also with sufficiently good approximate solu-

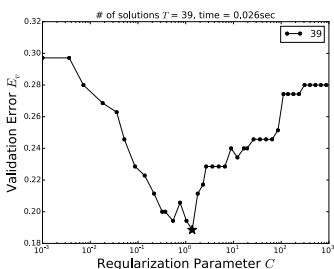

Figure 1: An illustration of the proposed framework. One of our algorithms presented in §4 automatically selected 39 regularization parameter values in $[10^{-3}, 10^3]$, and an upper bound of the validation error for each of them is obtained by solving an optimization problem approximately. Among those 39 values, the one with the smallest validation error upper bound (indicated as ★ at $C = 1.368$) is guaranteed to be $\varepsilon(= 0.1)$ *approximate regularization parameter* in the sense that the validation error for the regularization parameter is no greater by $\varepsilon$ than the smallest possible validation error in the whole interval $[10^{-3}, 10^3]$. See §5 for the setup (see also Figure 3 for the results with other options).

tions, which is computationally advantageous because completely solving an optimization problem is often much more costly than obtaining a reasonably good approximate solution.

Our main contribution in this paper is to show that a theoretically guaranteed choice of a regularization parameter in the above sense is possible with reasonable computational costs. To the best of our knowledge, there is no other existing methods for providing such a theoretical guarantee on CV error that can be used as generally as ours. Figure 1 illustrates the behavior of the algorithm for obtaining $\varepsilon = 0.1$ approximate regularization parameter (see §5 for the setup).

**Related works** *Optimal regularization parameter* can be found if its *exact regularization path* can be computed. Exact regularization path has been intensively studied [1, 2], but they are known to be numerically unstable and do not scale well. Furthermore, exact regularization path can be computed only for a limited class of problems whose solutions are written as piecewise-linear functions of the regularization parameter [3]. Our framework is much more efficient and can be applied to wider classes of problems whose exact regularization path cannot be computed. This work was motivated by recent studies on approximate regularization path [4, 5, 6, 7]. These approximate regularization paths have a property that the objective function value at each regularization parameter value is no greater by $\varepsilon$ than the optimal objective function value in the entire range of regularization parameters. Although these algorithms are much more stable and efficient than exact ones, for the task of tuning a regularization parameter, our interest is not in objective function values but in CV errors. Our approach is more suitable for regularization parameter tuning tasks in the sense that the approximation quality is guaranteed in terms of CV error.

As illustrated in Figure 1, we only compute a finite number of solutions, but still provide approximation guarantee in the whole interval of the regularization parameter. To ensure such a property, we need a novel CV error lower bound that is sufficiently tight and represented as a monotonic function of the regularization parameter. Although several CV error bounds (mostly for leave-one-out CV) of SVM and other similar learning frameworks exist (e.g., [8, 9, 10, 11]), none of them satisfy the above required properties. The idea of our CV error bound is inspired from recent studies on *safe screening* [12, 13, 14, 15, 16] (see Appendix A for the detail). Furthermore, we emphasize that our contribution is *not* in presenting a new generalization error bound, but in introducing a practical framework for providing a theoretical guarantee on the choice of a regularization parameter. Although generalization error bounds such as structural risk minimization [17] might be used for a rough tuning of a regularization parameter, they are known to be too loose to use as an alternative to CV (see, e.g., §11 in [18]). We also note that our contribution is *not* in presenting new method for regularization parameter tuning such as Bayesian optimization [19], random search [20] and gradient-based search [21]. As we demonstrate in experiments, our approach can provide a theoretical approximation guarantee of the regularization parameter selected by these existing methods.

## 2 Problem Setup

We consider linear binary classification problems. Let $\{(x_i, y_i) \in \mathbb{R}^d \times \{-1, 1\}\}_{i \in [n]}$ be the training set where $n$ is the size of the training set, $d$ is the input dimension, and $[n] := \{1, \ldots, n\}$. An independent held-out validation set with size $n'$ is denoted similarly as $\{(x_i', y_i') \in \mathbb{R}^d \times \{-1, 1\}\}_{i \in [n']}$. A linear decision function is written as $f(x) = w^\top x$, where $w \in \mathbb{R}^d$ is a vector of coefficients, and $^\top$ represents the transpose. We assume the availability of a held-out validation set only for simplifying the exposition. All the proposed methods presented in this paper can be straightforwardly

adapted to a cross-validation setup. Furthermore, the proposed methods can be *kernelized* if the loss function satisfies a certain condition. In this paper we focus on the following class of regularized convex loss minimization problems:

$$w_C^* := \arg \min_{w \in \mathbb{R}^d} \frac{1}{2} \|w\|^2 + C \sum_{i \in [n]} \ell(y_i, w^\top x_i), \tag{1}$$

where $C > 0$ is the regularization parameter, and $\| \cdot \|$ is the Euclidean norm. The loss function is denoted as $\ell : \{-1, 1\} \times \mathbb{R} \to \mathbb{R}$. We assume that $\ell(\cdot, \cdot)$ is convex and subdifferentiable in the 2nd argument. Examples of such loss functions include logistic loss, hinge loss, Huber-hinge loss, etc. For notational convenience, we denote the individual loss as $\ell_i(w) := \ell(y_i, w^\top x_i)$ for all $i \in [n]$. The optimal solution for the regularization parameter $C$ is explicitly denoted as $w_C^*$. We assume that the regularization parameter is defined in a finite interval $[C_\ell, C_u]$, e.g., $C_\ell = 10^{-3}$ and $C_u = 10^3$ as we did in the experiments.

For a solution $w \in \mathbb{R}^d$, the validation error[1] is defined as

$$E_v(w) := \frac{1}{n'} \sum_{i \in [n']} I(y_i' w^\top x_i' < 0), \tag{2}$$

where $I(\cdot)$ is the indicator function. In this paper, we consider two problem setups. The first problem setup is, given a set of (either optimal or approximate) solutions $w_{C_1}^*, \ldots, w_{C_T}^*$ at $T$ different regularization parameters $C_1, \ldots, C_T \in [C_\ell, C_u]$, to compute the approximation level $\varepsilon$ such that

$$\min_{C_t \in \{C_1, \ldots, C_T\}} E_v(w_{C_t}^*) - E_v^* \leq \varepsilon, \quad \text{where} \quad E_v^* := \min_{C \in [C_l, C_u]} E_v(w_C^*), \tag{3}$$

by which we can find how accurate our search (grid-search, typically) is in a sense of the deviation of the achieved validation error from the true minimum in the range, i.e., $E_v^*$. The second problem setup is, given the approximation level $\varepsilon$, to find an $\varepsilon$-approximate regularization parameter within an interval $C \in [C_l, C_u]$, which is defined as an element of the following set

$$\mathcal{C}(\varepsilon) := \left\{ C \in [C_l, C_u] \,\middle|\, E_v(w_C^*) - E_v^* \leq \varepsilon \right\}.$$

Our goal in this second setup is to derive an efficient exploration procedure which achieves the specified validation approximation level $\varepsilon$. These two problem setups are both common scenarios in practical data analysis, and can be solved by using our proposed framework for computing a path of validation error lower bounds.

## 3 Validation error lower bounds as a function of regularization parameter

In this section, we derive a validation error lower bound which is represented as a function of the regularization parameter $C$. Our basic idea is to compute a lower and an upper bound of the inner product score $w_C^{*\top} x_i'$ for each validation input $x_i', i \in [n']$, as a function of the regularization parameter $C$. For computing the bounds of $w_C^{*\top} x_i'$, we use a solution (either optimal or approximate) for a different regularization parameter $\tilde{C} \neq C$.

### 3.1 Score bounds

We first describe how to obtain a lower and an upper bound of inner product score $w_C^{*\top} x_i'$ based on an approximate solution $\hat{w}_{\tilde{C}}$ at a different regularization parameter $\tilde{C} \neq C$.

**Lemma 1.** *Let $\hat{w}_{\tilde{C}}$ be an approximate solution of the problem (1) for a regularization parameter value $\tilde{C}$ and $\xi_i(\hat{w}_C)$ be a subgradient of $\ell_i$ at $w = \hat{w}_C$ such that a subgradient of the objective function is*

$$g(\hat{w}_{\tilde{C}}) := \hat{w}_C + \tilde{C} \sum_{i \in [n]} \xi_i(\hat{w}_C). \tag{4}$$

Then, for any $C > 0$, the score $w_C^{*\top} x_i', i \in [n']$, satisfies

$$w_C^{*\top} x_i' \geq LB(w_C^{*\top} x_i' | \hat{w}_{\tilde{C}}) := \begin{cases} \alpha(\hat{w}_{\tilde{C}}, x_i') - \frac{1}{\tilde{C}}(\beta(\hat{w}_{\tilde{C}}, x_i') + \gamma(g(\hat{w}_{\tilde{C}}), x_i'))C, \text{if } C > \tilde{C}, \\ -\beta(\hat{w}_{\tilde{C}}, x_i') + \frac{1}{\tilde{C}}(\alpha(\hat{w}_{\tilde{C}}, x_i') + \delta(g(\hat{w}_{\tilde{C}}), x_i'))C, \text{if } C < \tilde{C}, \end{cases} \quad (5a)$$

$$w_C^{*\top} x_i' \leq UB(w_C^{*\top} x_i' | \hat{w}_{\tilde{C}}) := \begin{cases} -\beta(\hat{w}_{\tilde{C}}, x_i') + \frac{1}{\tilde{C}}(\alpha(\hat{w}_{\tilde{C}}, x_i') + \delta(g(\hat{w}_{\tilde{C}}), x_i'))C, \text{if } C > \tilde{C}, \\ \alpha(\hat{w}_{\tilde{C}}, x_i') - \frac{1}{\tilde{C}}(\beta(\hat{w}_{\tilde{C}}, x_i') + \gamma(g(\hat{w}_{\tilde{C}}), x_i'))C, \text{if } C < \tilde{C}, \end{cases} \quad (5b)$$

where

$$\alpha(w_{\tilde{C}}^*, x_i') := \frac{1}{2}(\|w_{\tilde{C}}^*\| \|x_i'\| + w_{\tilde{C}}^{*\top} x_i') \geq 0, \quad \gamma(g(\hat{w}_{\tilde{C}}), x_i') := \frac{1}{2}(\|g(\hat{w}_{\tilde{C}})\| \|x_i'\| + g(\hat{w}_{\tilde{C}})^\top x_i') \geq 0,$$

$$\beta(w_{\tilde{C}}^*, x_i') := \frac{1}{2}(\|w_{\tilde{C}}^*\| \|x_i'\| - w_{\tilde{C}}^{*\top} x_i') \geq 0, \quad \delta(g(\hat{w}_{\tilde{C}}), x_i') := \frac{1}{2}(\|g(\hat{w}_{\tilde{C}})\| \|x_i'\| - g(w_{\tilde{C}})^\top x_i') \geq 0.$$

The proof is presented in Appendix A. Lemma 1 tells that we have a lower and an upper bound of the score $w_C^{*\top} x_i'$ for each validation instance that linearly change with the regularization parameter $C$. When $\hat{w}_{\tilde{C}}$ is optimal, it can be shown that (see Proposition B.24 in [22]) there exists a subgradient such that $g(\hat{w}_{\tilde{C}}) = 0$, meaning that the bounds are tight because $\gamma(g(\hat{w}_{\tilde{C}}), x_i') = \delta(g(\hat{w}_{\tilde{C}}), x_i') = 0$.

**Corollary 2.** *When $C = \tilde{C}$, the score $w_{\tilde{C}}^{*\top} x_i', i \in [n']$, for the regularization parameter value $\tilde{C}$ itself satisfies*

$$w_{\tilde{C}}^{*\top} x_i' \geq LB(w_{\tilde{C}}^{*\top} x_i' | \hat{w}_{\tilde{C}}) = \hat{w}_{\tilde{C}}^\top x_i' - \gamma(g(\hat{w}_{\tilde{C}}), x_i'), \quad w_{\tilde{C}}^{*\top} x_i' \leq UB(w_{\tilde{C}}^{*\top} x_i' | \hat{w}_{\tilde{C}}) = \hat{w}_{\tilde{C}}^\top x_i' + \delta(g(\hat{w}_{\tilde{C}}), x_i').$$

The results in Corollary 2 are obtained by simply substituting $C = \tilde{C}$ into (5a) and (5b).

### 3.2 Validation Error Bounds

Given a lower and an upper bound of the score of each validation instance, a lower bound of the validation error can be computed by simply using the following facts:

$$y_i' = +1 \text{ and } UB(w_C^{*\top} x_i' | \hat{w}_{\tilde{C}}) < 0 \Rightarrow \text{mis-classified}, \quad (6a)$$

$$y_i' = -1 \text{ and } LB(w_C^{*\top} x_i' | \hat{w}_{\tilde{C}}) > 0 \Rightarrow \text{mis-classified}. \quad (6b)$$

Furthermore, since the bounds in Lemma 1 linearly change with the regularization parameter $C$, we can identify the interval of $C$ within which the validation instance is guaranteed to be mis-classified.

**Lemma 3.** *For a validation instance with $y_i' = +1$, if*

$$\tilde{C} < C < \frac{\beta(\hat{w}_{\tilde{C}}, x_i')}{\alpha(\hat{w}_{\tilde{C}}, x_i') + \delta(g(\hat{w}_{\tilde{C}}), x_i')} \tilde{C} \quad \text{or} \quad \frac{\alpha(\hat{w}_{\tilde{C}}, x_i')}{\beta(\hat{w}_{\tilde{C}}, x_i') + \gamma(g(\hat{w}_{\tilde{C}}), x_i')} \tilde{C} < C < \tilde{C},$$

*then the validation instance $(x_i', y_i')$ is mis-classified. Similarly, for a validation instance with $y_i' = -1$, if*

$$\tilde{C} < C < \frac{\alpha(\hat{w}_{\tilde{C}}, x_i')}{\beta(\hat{w}_{\tilde{C}}, x_i') + \gamma(g(\hat{w}_{\tilde{C}}), x_i')} \tilde{C} \quad \text{or} \quad \frac{\beta(\hat{w}_{\tilde{C}}, x_i')}{\alpha(\hat{w}_{\tilde{C}}, x_i') + \delta(g(\hat{w}_{\tilde{C}}), x_i')} \tilde{C} < C < \tilde{C},$$

*then the validation instance $(x_i', y_i')$ is mis-classified.*

This lemma can be easily shown by applying (5) to (6).

As a direct consequence of Lemma 3, the lower bound of the validation error is represented as a function of the regularization parameter $C$ in the following form.

**Theorem 4.** *Using an approximate solution $\hat{w}_{\tilde{C}}$ for a regularization parameter $\tilde{C}$, the validation error $E_v(w_C^*)$ for any $C > 0$ satisfies*

$$E_v(w_C^*) \geq LB(E_v(w_C^*) | \hat{w}_{\tilde{C}}) := \quad (7)$$

$$\frac{1}{n'} \left( \sum_{y_i'=+1} I\left(\tilde{C} < C < \frac{\beta(\hat{w}_{\tilde{C}}, x_i')}{\alpha(\hat{w}_{\tilde{C}}, x_i') + \delta(g(\hat{w}_{\tilde{C}}), x_i')} \tilde{C}\right) + \sum_{y_i'=+1} I\left(\frac{\alpha(\hat{w}_{\tilde{C}}, x_i')}{\beta(\hat{w}_{\tilde{C}}, x_i') + \gamma(g(\hat{w}_{\tilde{C}}), x_i')} \tilde{C} < C < \tilde{C}\right) \right.$$

$$\left. + \sum_{y_i'=-1} I\left(\tilde{C} < C < \frac{\alpha(\hat{w}_{\tilde{C}}, x_i')}{\beta(\hat{w}_{\tilde{C}}, x_i') + \gamma(g(\hat{w}_{\tilde{C}}), x_i')} \tilde{C}\right) + \sum_{y_i'=-1} I\left(\frac{\beta(\hat{w}_{\tilde{C}}, x_i')}{\alpha(\hat{w}_{\tilde{C}}, x_i') + \delta(g(\hat{w}_{\tilde{C}}), x_i')} \tilde{C} < C < \tilde{C}\right) \right).$$

---

Algorithm 1: Computing the approximation level $\varepsilon$ from the given set of solutions

---

**Input:** $\{(x_i, y_i)\}_{i \in [n]}, \{(x_i', y_i')\}_{i \in [n']}, C_l, C_u, \mathcal{W} := \{w_{\tilde{C}_1}, \ldots, w_{\tilde{C}_T}\}$

  1: $E_v^{\mathrm{best}} \leftarrow \min_{\tilde{C}_t \in \{\tilde{C}_1, \ldots, \tilde{C}_T\}} UB(E_v(w_{\tilde{C}_t}^*) | w_{\tilde{C}_t})$

  2: $LB(E_v^*) \leftarrow \min_{c \in [C_l, C_u]} \left\{ \max_{\tilde{C}_t \in \{\tilde{C}_1, \ldots, \tilde{C}_T\}} LB(E_v(w_c^*) | w_{\tilde{C}_t}) \right\}$

**Output:** $\varepsilon = E_v^{\mathrm{best}} - LB(E_v^*)$

---

The lower bound (7) is a staircase function of the regularization parameter $C$.

**Remark 5.** *We note that our validation error lower bound is inspired from recent studies on safe screening [12, 13, 14, 15, 16], which identifies sparsity of the optimal solutions before solving the optimization problem. A key technique used in those studies is to bound Lagrange multipliers at the optimal, and we utilize this technique to prove Lemma 1, which is a core of our framework.*

By setting $C = \tilde{C}$, we can obtain a lower and an upper bound of the validation error for the regularization parameter $\tilde{C}$ itself, which are used in the algorithm as a stopping criteria for obtaining an approximate solution $\hat{w}_{\tilde{C}}$.

**Corollary 6.** *Given an approximate solution $\hat{w}_{\tilde{C}}$, the validation error $E_v(w_{\tilde{C}}^*)$ satisfies*

$$
E_v(w_{\tilde{C}}^*) \geq LB(E_v(w_{\tilde{C}}^*) | \hat{w}_{\tilde{C}})
$$

$$
= \frac{1}{n'} \left( \sum_{y_i'=+1} I\big( \hat{w}_{\tilde{C}}^\top x_i' + \delta(g(\hat{w}_{\tilde{C}}), x_i') < 0 \big) + \sum_{y_i'=-1} I\big( \hat{w}_{\tilde{C}}^\top x_i' - \gamma(g(\hat{w}_{\tilde{C}}), x_i') > 0 \big) \right), \tag{8a}
$$

$$
E_v(w_{\tilde{C}}^*) \leq UB(E_v(w_{\tilde{C}}^*) | \hat{w}_{\tilde{C}})
$$

$$
= 1 - \frac{1}{n'} \left( \sum_{y_i'=+1} I\big( \hat{w}_{\tilde{C}}^\top x_i' - \gamma(g(\hat{w}_{\tilde{C}}), x_i') \geq 0 \big) + \sum_{y_i'=-1} I\big( \hat{w}_{\tilde{C}}^\top x_i' + \delta(g(\hat{w}_{\tilde{C}}), x_i') \leq 0 \big) \right). \tag{8b}
$$

## 4 Algorithm

In this section we present two algorithms for each of the two problems discussed in §2. Due to the space limitation, we roughly describe the most fundamental forms of these algorithms. Details and several extensions of the algorithms are presented in supplementary appendices B and C.

### 4.1 Problem setup 1: Computing the approximation level $\varepsilon$ from a given set of solutions

Given a set of (either optimal or approximate) solutions $\hat{w}_{\tilde{C}_1}, \ldots, \hat{w}_{\tilde{C}_T}$, obtained e.g., by ordinary grid-search, our first problem is to provide a theoretical approximation level $\varepsilon$ in the sense of (3)[2]. This problem can be solved easily by using the validation error lower bounds developed in §3.2. The algorithm is presented in Algorithm 1, where we compute the current best validation error $E_v^{\mathrm{best}}$ in line 1, and a lower bound of the best possible validation error $E_v^* := \min_{C \in [C_\ell, C_u]} E_v(w_C^*)$ in line 2. Then, the approximation level $\varepsilon$ can be simply obtained by subtracting the latter from the former. We note that $LB(E_v^*)$, the lower bound of $E_v^*$, can be easily computed by using $T$ evaluation error lower bounds $LB(E_v(w_C^*) | w_{\tilde{C}_t})$, $t = 1, \ldots, T$, because they are staircase functions of $C$.

### 4.2 Problem setup 2: Finding an $\varepsilon$-approximate regularization parameter

Given a desired approximation level $\varepsilon$ such as $\varepsilon = 0.01$, our second problem is to find an $\varepsilon$-approximate regularization parameter. To this end we develop an algorithm that produces a set of optimal or approximate solutions $\hat{w}_{\tilde{C}_1}, \ldots, \hat{w}_{\tilde{C}_T}$ such that, if we apply Algorithm 1 to this sequence, then approximation level would be smaller than or equal to $\varepsilon$. Algorithm 2 is the pseudo-code of this algorithm. It computes approximate solutions for an increasing sequence of regularization parameters in the main loop (lines 2-11).

**Algorithm 2:** Finding an $\varepsilon$ approximate regularization parameter with approximate solutions

---

**Input:** $\{(x_i, y_i)\}_{i\in[n]}, \{(x_i', y_i')\}_{i\in[n']}, C_l, C_u, \varepsilon$

1: $t \leftarrow 1, \tilde{C}_t \leftarrow C_l, C^{\text{best}} \leftarrow C_l, E_v^{\text{best}} \leftarrow 1$
2: **while** $\tilde{C}_t \leq C_u$ **do**
3:    $\hat{w}_{\tilde{C}_t} \leftarrow$ solve (1) approximately for $C = \tilde{C}_t$
4:    Compute $UB(E_v(w_{\tilde{C}_t}^*)|\hat{w}_{\tilde{C}_t})$ by (8b).
5:    **if** $UB(E_v(w_{\tilde{C}_t}^*)|\hat{w}_{\tilde{C}_t}) < E_v^{\text{best}}$ **then**
6:       $E_v^{\text{best}} \leftarrow UB(E_v(w_{\tilde{C}_t}^*)|\hat{w}_{\tilde{C}_t})$
7:       $C^{\text{best}} \leftarrow \tilde{C}_t$
8:    **end if**
9:    Set $\tilde{C}_{t+1}$ by (10)
10:   $t \leftarrow t+1$
11: **end while**
**Output:** $C^{\text{best}} \in \mathcal{C}(\varepsilon)$.

---

Let us now consider $t^{\text{th}}$ iteration in the main loop, where we have already computed $t-1$ approximate solutions $\hat{w}_{\tilde{C}_1}, \ldots, \hat{w}_{\tilde{C}_{t-1}}$ for $\tilde{C}_1 < \ldots < \tilde{C}_{t-1}$. At this point,

$$C^{\text{best}} := \arg\min_{\tilde{C}_\tau \in \{\tilde{C}_1,\ldots,\tilde{C}_{t-1}\}} UB(E_v(w_{\tilde{C}_\tau}^*)|\hat{w}_{\tilde{C}_\tau}),$$

is the best (in worst-case) regularization parameter obtained so far and it is guaranteed to be an $\varepsilon$-approximate regularization parameter in the interval $[C_l, \tilde{C}_t]$ in the sense that the validation error,

$$E_v^{\text{best}} := \min_{\tilde{C}_\tau \in \{\tilde{C}_1,\ldots,\tilde{C}_{t-1}\}} UB(E_v(w_{\tilde{C}_\tau}^*)|\hat{w}_{\tilde{C}_\tau}),$$

is shown to be at most greater by $\varepsilon$ than the smallest possible validation error in the interval $[C_l, \tilde{C}_t]$. However, we are not sure whether $C^{\text{best}}$ can still keep $\varepsilon$-approximation property for $C > \tilde{C}_t$. Thus, in line 3, we approximately solve the optimization problem (1) at $C = \tilde{C}_t$ and obtain an approximate solution $\hat{w}_{\tilde{C}_t}$. Note that the approximate solution $\hat{w}_{\tilde{C}_t}$ must be sufficiently good enough in the sense that $UB(E_v(w_{\tilde{C}_\tau}^*)|\hat{w}_{\tilde{C}_\tau}) - LB(E_v(w_{\tilde{C}_\tau}^*)|\hat{w}_{\tilde{C}_\tau})$ is sufficiently smaller than $\varepsilon$ (typically $0.1\varepsilon$). If the upper bound of the validation error $UB(E_v(w_{\tilde{C}_\tau}^*)|\hat{w}_{\tilde{C}_\tau})$ is smaller than $E_v^{\text{best}}$, we update $E_v^{\text{best}}$ and $C^{\text{best}}$ (lines 5-8).

Our next task is to find $\tilde{C}_{t+1}$ in such a way that $C^{\text{best}}$ is an $\varepsilon$-approximate regularization parameter in the interval $[C_l, \tilde{C}_{t+1}]$. Using the validation error lower bound in Theorem 4, the task is to find the smallest $\tilde{C}_{t+1} > \tilde{C}_t$ that violates

$$E_v^{\text{best}} - LB(E_v(w_C^*)|\hat{w}_{\tilde{C}_t}) \leq \varepsilon, \quad \forall C \in [\tilde{C}_t, C_u], \tag{9}$$

In order to formulate such a $\tilde{C}_{t+1}$, let us define

$$\mathcal{P} := \{i \in [n'] | y_i' = +1, UB(w_{\tilde{C}_t}^{*\top} x_i'|\hat{w}_{\tilde{C}_t}) < 0\}, \mathcal{N} := \{i \in [n'] | y_i' = -1, LB(w_{\tilde{C}_t}^{*\top} x_i'|\hat{w}_{\tilde{C}_t}) > 0\}.$$

Furthermore, let

$$\Gamma := \left\{ \frac{\beta(\hat{w}_{\tilde{C}_t}, x_i')}{\alpha(\hat{w}_{\tilde{C}_t}, x_i') + \delta(g(\hat{w}_{\tilde{C}_t}), x_i')} \tilde{C}_t \right\}_{i\in\mathcal{P}} \cup \left\{ \frac{\alpha(\hat{w}_{\tilde{C}_t}, x_i')}{\beta(\hat{w}_{\tilde{C}_t}, x_i') + \gamma(g(\hat{w}_{\tilde{C}_t}), x_i')} \tilde{C}_t \right\}_{i\in\mathcal{N}},$$

and denote the $k^{\text{th}}$-smallest element of $\Gamma$ as $k^{\text{th}}(\Gamma)$ for any natural number $k$. Then, the smallest $\tilde{C}_{t+1} > \tilde{C}_t$ that violates (9) is given as

$$\tilde{C}_{t+1} \leftarrow (\lfloor n'(LB(E_v(w_{\tilde{C}_t}^*)|\hat{w}_{\tilde{C}_t}) - E_v^{\text{best}} + \varepsilon) \rfloor + 1)^{\text{th}}(\Gamma). \tag{10}$$

## 5 Experiments

In this section we present experiments for illustrating the proposed methods. Table 2 summarizes the datasets used in the experiments. They are taken from libsvm dataset repository [23]. All the input features except D9 and D10 were standardized to $[-1, 1]^3$. For illustrative results, the instances were randomly divided into a training and a validation sets in roughly equal sizes. For quantitative results, we used 10-fold CV. We used Huber hinge loss (e.g., [24]) which is convex and subdifferentiable with respect to the second argument. The proposed methods are free from the choice of optimization solvers. In the experiments, we used an optimization solver described in [25], which is also implemented in well-known *liblinear* software [26]. Our slightly modified code

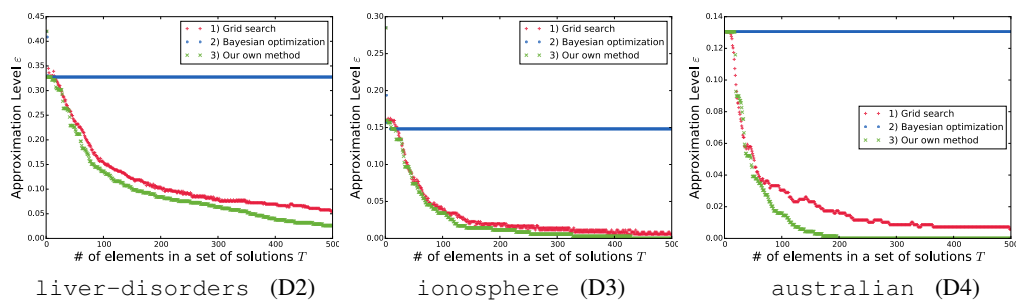

liver-disorders (D2)    ionosphere (D3)    australian (D4)

Figure 2: Illustrations of Algorithm 1 on three benchmark datasets (D2, D3, D4). The plots indicate how the approximation level $\varepsilon$ improves as the number of solutions $T$ increases in grid-search (red), Bayesian optimization (blue) and our own method (green, see the main text).

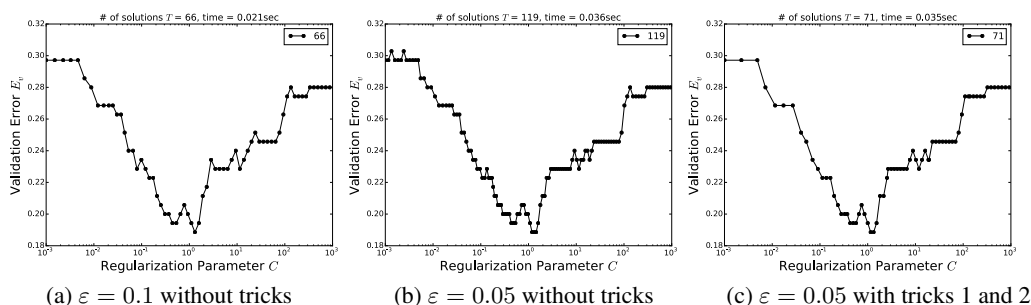

(a) $\varepsilon = 0.1$ without tricks    (b) $\varepsilon = 0.05$ without tricks    (c) $\varepsilon = 0.05$ with tricks 1 and 2

Figure 3: Illustrations of Algorithm 2 on `ionosphere` (D3) dataset for (a) **op2** with $\varepsilon = 0.10$, (b) **op2** with $\varepsilon = 0.05$ and (c) **op3** with $\varepsilon = 0.05$, respectively. Figure 1 also shows the result for **op3** with $\varepsilon = 0.10$.

(for adaptation to Huber hinge loss) is provided as a supplementary material, and is also available on `https://github.com/takeuchi-lab/RPCVELB`. Whenever possible, we used *warm-start* approach, i.e., when we trained a new solution, we used the closest solutions trained so far (either approximate or optimal ones) as the initial starting point of the optimizer. All the computations were conducted by using a single core of an HP workstation Z800 (Xeon(R) CPU X5675 (3.07GHz), 48GB MEM). In all the experiments, we set $C_\ell = 10^{-3}$ and $C_u = 10^3$.

**Results on problem 1**  We applied Algorithm 1 in §4 to a set of solutions obtained by 1) grid-search, 2) Bayesian optimization (BO) with expected improvement acquisition function, and 3) adaptive search with our framework which sequentially computes a solution whose validation lower bound is smallest based on the information obtained so far. Figure 2 illustrates the results on three datasets, where we see how the approximation level $\varepsilon$ in the vertical axis changes as the number of solutions ($T$ in our notation) increases. In grid-search, as we increase the grid points, the approximation level $\varepsilon$ tends to be improved. Since BO tends to focus on a small region of the regularization parameter, it was difficult to tightly bound the approximation level. We see that the adaptive search, using our framework straightforwardly, seems to offer slight improvement from grid-search.

**Results on problem 2**  We applied Algorithm 2 to benchmark datasets for demonstrating theoretically guaranteed choice of a regularization parameter is possible with reasonable computational costs. Besides the algorithm presented in §4, we also tested a variant described in supplementary Appendix B. Specifically, we have three algorithm options. In the first option (**op1**), we used optimal solutions $\{w^*_{\tilde{C}_t}\}_{t \in [T]}$ for computing CV error lower bounds. In the second option (**op2**), we instead used approximate solutions $\{\hat{w}_{\tilde{C}_t}\}_{t \in [T]}$. In the last option (**op3**), we additionally used speed-up tricks described in supplementary Appendix B. We considered four different choices of $\varepsilon \in \{0.1, 0.05, 0.01, 0\}$. Note that $\varepsilon = 0$ indicates the task of finding the exactly optimal regular-

Table 1: Computational costs. For each of the three options and $\varepsilon \in \{0.10, 0.05, 0.01, 0\}$, the number of optimization problems solved (denoted as $T$) and the total computational costs (denoted as time) are listed. Note that, for **op2**, there are no results for $\varepsilon = 0$.

| $\varepsilon$ | | op1 (using $w^*_{\tilde{C}}$) $T$ | time (sec) | op2 (using $\hat{w}_{\tilde{C}}$) $T$ | time (sec) | op3 (using tricks) $T$ | time (sec) | | op1 (using $w^*_{\tilde{C}}$) $T$ | time (sec) | op2 (using $\hat{w}_{\tilde{C}}$) $T$ | time (sec) | op3 (using tricks) $T$ | time (sec) |
|---|---|---|---|---|---|---|---|---|---|---|---|---|---|---|
| 0.10 | D1 | 30 | 0.068 | 32 | 0.031 | 33 | 0.041 | D6 | 92 | 1.916 | 93 | 0.975 | 62 | 0.628 |
| 0.05 | | 68 | 0.124 | 70 | 0.061 | 57 | 0.057 | | 207 | 4.099 | 209 | 2.065 | 123 | 1.136 |
| 0.01 | | 234 | 0.428 | 324 | 0.194 | 205 | 0.157 | | 1042 | 16.31 | 1069 | 9.686 | 728 | 5.362 |
| 0 | | 442 | 0.697 | N.A. | | 383 | 0.629 | | 4276 | 57.57 | N.A. | | 2840 | 44.68 |
| 0.10 | D2 | 221 | 0.177 | 223 | 0.124 | 131 | 0.084 | D7 | 289 | 8.492 | 293 | 5.278 | 167 | 3.319 |
| 0.05 | | 534 | 0.385 | 540 | 0.290 | 367 | 0.218 | | 601 | 16.18 | 605 | 9.806 | 379 | 6.604 |
| 0.01 | | 1503 | 0.916 | 2183 | 0.825 | 1239 | 0.623 | | 2532 | 57.79 | 2788 | 35.21 | 1735 | 24.04 |
| 0 | | 10939 | 6.387 | N.A. | | 6275 | 3.805 | | 67490 | 1135 | N.A. | | 42135 | 760.8 |
| 0.10 | D3 | 61 | 0.617 | 62 | 0.266 | 43 | 0.277 | D8 | 72 | 0.761 | 74 | 0.604 | 66 | 0.606 |
| 0.05 | | 123 | 1.073 | 129 | 0.468 | 73 | 0.359 | | 192 | 1.687 | 195 | 1.162 | 110 | 0.926 |
| 0.01 | | 600 | 4.776 | 778 | 0.716 | 270 | 0.940 | | 1063 | 8.257 | 1065 | 6.238 | 614 | 4.043 |
| 0 | | 5412 | 26.39 | N.A. | | 815 | 6.344 | | 34920 | 218.4 | N.A. | | 15218 | 99.57 |
| 0.10 | D4 | 27 | 0.169 | 27 | 0.088 | 23 | 0.093 | D9 | 134 | 360.2 | 136 | 201.0 | 89 | 74.37 |
| 0.05 | | 64 | 0.342 | 65 | 0.173 | 47 | 0.153 | | 317 | 569.9 | 323 | 280.7 | 200 | 128.5 |
| 0.01 | | 167 | 0.786 | 181 | 0.418 | 156 | 0.399 | | 1791 | 2901 | 1822 | 1345 | 1164 | 657.4 |
| 0 | | 342 | 1.317 | N.A. | | 345 | 1.205 | | 85427 | 106937 | N.A. | | 63300 | 98631 |
| 0.10 | D5 | 62 | 0.236 | 63 | 0.108 | 45 | 0.091 | D10 | $E_v < 0.10$ | | $E_v < 0.10$ | | $E_v < 0.10$ | |
| 0.05 | | 108 | 0.417 | 109 | 0.171 | 77 | 0.137 | | $E_v < 0.05$ | | $E_v < 0.05$ | | $E_v < 0.05$ | |
| 0.01 | | 421 | 1.201 | 440 | 0.631 | 258 | 0.401 | | 157 | 81.75 | 162 | 31.02 | 114 | 36.81 |
| 0 | | 2330 | 4.540 | N.A. | | 968 | 2.451 | | 258552 | 85610 | N.A. | | 42040 | 23316 |

ization parameter. In some datasets, the smallest validation errors are less than 0.1 or 0.05, in which cases we do not report the results (indicated as "$E_v < 0.05$" etc.). In trick1, we initially computed solutions at four different regularization parameter values evenly allocated in $[10^{-3}, 10^3]$ in the logarithmic scale. In trick2, the next regularization parameter $\tilde{C}_{t+1}$ was set by replacing $\varepsilon$ in (10) with $1.5\varepsilon$ (see supplementary Appendix B). For the purpose of illustration, we plot examples of validation error curves in several setups. Figure 3 shows the validation error curves of `ionosphere` (D3) dataset for several options and $\varepsilon$.

Table 1 shows the number of optimization problems solved in the algorithm (denoted as $T$), and the total computation time in CV setups. The computational costs mostly depend on $T$, which gets smaller as $\varepsilon$ increases. Two tricks in supplementary Appendix B was effective in most cases for reducing $T$. In addition, we see the advantage of using approximate solutions by comparing the computation times of **op1** and **op2** (though this strategy is only for $\varepsilon \neq 0$). Overall, the results suggest that the proposed algorithm allows us to find theoretically guaranteed approximate regularization parameters with reasonable costs except for $\varepsilon = 0$ cases. For example, the algorithm found an $\varepsilon = 0.01$ approximate regularization parameter within a minute in 10-fold CV for a dataset with more than 50000 instances (see the results on D10 for $\varepsilon = 0.01$ with **op2** and **op3** in Table 1).

Table 2: Benchmark datasets used in the experiments.

| | dataset name | sample size | input dimension | | | dataset name | sample size | input dimension |
|---|---|---|---|---|---|---|---|---|
| D1 | heart | 270 | 13 | D6 | | german.numer | 1000 | 24 |
| D2 | liver-disorders | 345 | 6 | D7 | | svmguide3 | 1284 | 21 |
| D3 | ionosphere | 351 | 34 | D8 | | svmguide1 | 7089 | 4 |
| D4 | australian | 690 | 14 | D9 | | a1a | 32561 | 123 |
| D5 | diabetes | 768 | 8 | D10 | | w8a | 64700 | 300 |

## 6 Conclusions and future works

We presented a novel algorithmic framework for computing CV error lower bounds as a function of the regularization parameter. The proposed framework can be used for a theoretically guaranteed choice of a regularization parameter. Additional advantage of this framework is that we only need to compute a set of sufficiently good approximate solutions for obtaining such a theoretical guarantee, which is computationally advantageous. As demonstrated in the experiments, our algorithm is practical in the sense that the computational cost is reasonable as long as the approximation quality $\varepsilon$ is not too close to 0. An important future work is to extend the approach to multiple hyper-parameters tuning setups.

## Footnotes

[1] For simplicity, we regard a validation instance whose score is exactly zero, i.e., $w^\top x_i' = 0$, is correctly classified in (2). Hereafter, we assume that there are no validation instances whose input vector is completely 0, i.e., $x_i' = 0$, because those instances are always correctly classified according to the definition in (2).

[2] When we only have approximate solutions $\hat{w}_{\tilde{C}_1}, \ldots, \hat{w}_{\tilde{C}_T}$, Eq. (3) is slightly incorrect. The first term of the l.h.s. of (3) should be $\min_{\tilde{C}_t \in \{\tilde{C}_1, \ldots, \tilde{C}_T\}} UB(E_v(\hat{w}_{\tilde{C}_t}) | \hat{w}_{\tilde{C}_t})$.

[3] We use D9 and D10 as they are for exploiting sparsity.

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
