[Supplementary Material · appendix.pdf]

# A Proof of Lemma 1

In this section we prove Lemma 1. First we present two propositions which are used of proving Lemma 1.

**Proposition 7.** *Consider the following general problem:*

$$\min_{z} \ \phi(z) \quad \text{s.t. } z \in \mathcal{Z}, \tag{11}$$

*where $\phi : \mathcal{Z} \to \mathbb{R}$ is a subdifferentiable convex function and $\mathcal{Z} \subset \mathbb{R}^d$ is a convex set. Then a solution $z^*$ is the optimal solution of (11) if and only if there exists a subgradient $\xi \in \partial\phi(z^*)$ such that*

$$\xi^\top (z^* - z) \leq 0, \quad \forall z \in \mathcal{Z},$$

*where $\partial\phi(z^*)$ is the set of all subgradients of convex function $\phi$ at $z = z^*$.*

See, for example, Proposition B.24 in [22] for the proof of Proposition 7.

**Proposition 8.** *Let $p, q \in \mathbb{R}^d$ be arbitrary $d$-dimensional vectors and $r > 0$ be an arbitrary positive constant. Then, the solutions of the following optimization problem can be explicitly obtained as follows:*

$$p^\top q - \|p\|r = \min_{z \in \mathbb{R}^d} \ p^\top z \quad \text{s.t. } \|z - q\|^2 \leq r^2, \tag{12}$$

$$p^\top q + \|p\|r = \max_{z \in \mathbb{R}^d} \ p^\top z \quad \text{s.t. } \|z - q\|^2 \leq r^2. \tag{13}$$

*Proof of Proposition 8.* Using a Lagrange multiplier $\lambda > 0$, the problem (12) is rewritten as

$$\begin{aligned}
&\min_{z \in \mathbb{R}^d} \ p^\top z \quad \text{s.t. } \|z - q\|^2 \leq r^2 \\
&= \min_{z \in \mathbb{R}^d} \max_{\lambda > 0} \left( p^\top z + \lambda(\|z - q\|^2 \leq r^2) \right) \\
&= \max_{\lambda > 0} \left( -\lambda r^2 + \min_{z} \left( \lambda\|z - p\|^2 + p^\top z \right) \right) \\
&= \max_{\lambda > 0} \ H(\lambda) := \left( -\lambda r^2 - \frac{\|p\|^2}{4\lambda} + p^\top q \right),
\end{aligned}$$

where $\lambda$ is strictly positive because the constraint $\|p - q\|^2 \leq r^2$ is strictly active at the optimal solution. By letting $\partial H(\lambda)/\partial\lambda = 0$, the optimal $\lambda$ is written as

$$\lambda^* := \frac{\|p\|}{2r} = \arg\max_{\lambda > 0} \ H(\lambda).$$

Substituting $\lambda^*$ into $H(\lambda)$,

$$p^\top q - \|p\|r = \max_{\lambda > 0} \ H(\lambda).$$

The upper bound of $p^\top z$ in (13) can be shown similarly. ∎

*Proof of Lemma 1.* From Proposition 7, the optimal solution $w_C^*$ satisfies

$$\left( w_C^* + C \sum_{i \in [n]} \xi_i(w_C^*) \right)^\top (w_C^* - \hat{w}_{\tilde{C}}) \leq 0, \tag{14}$$

where $\xi_i(w_C^*)$ is a subgradient of $\ell_i$ at $w = w_C^*$ for any $i \in [n]$ .

Since from $\ell_i$ is convex for any $i \in [n]$ and the definition of a subgradient, we have the following two inequalities:

$$\ell_i(w_C^*) \geq \ell_i(\hat{w}_{\tilde{C}}) + \xi_i(\hat{w}_{\tilde{C}})^\top (w_C^* - \hat{w}_{\tilde{C}}).$$
$$\ell_i(\hat{w}_{\tilde{C}}) \geq \ell_i(w_C^*) + \xi_i(w_C^*)^\top (\hat{w}_{\tilde{C}} - w_C^*).$$

Combining these two inequalities, we have

$$\xi_i(w_C^*)^\top(w_C^* - \hat{w}_{\tilde{C}}) \geq \xi_i(\hat{w}_{\tilde{C}})^\top(w_C^* - \hat{w}_{\tilde{C}}). \tag{15}$$

Substituting (15) into (14),

$$w_C^{*\top}(w_C^* - \hat{w}_{\tilde{C}}) + C\sum_{i\in[n]}\xi_i(\hat{w}_{\tilde{C}})^\top(w_C^* - \hat{w}_{\tilde{C}}) \leq 0. \tag{16}$$

From (4),

$$\sum_{i\in[n]}\xi_i(\hat{w}_{\tilde{C}}) = \frac{1}{\tilde{C}}\Big(g(\hat{w}_{\tilde{C}}) - \hat{w}_{\tilde{C}}\Big). \tag{17}$$

Substituting (17) into (16),

$$w_C^{*\top}(w_C^* - \hat{w}_{\tilde{C}}) + \frac{C}{\tilde{C}}\Big(g(\hat{w}_{\tilde{C}}) - \hat{w}_{\tilde{C}}\Big)^\top(w_C^* - \hat{w}_{\tilde{C}}) \leq 0$$

$$\Leftrightarrow \left\|w_C^* - \frac{1}{2}\Big(\hat{w} - \frac{C}{\tilde{C}}(g(\hat{w}) - \hat{w})\Big)\right\|^2 \leq \left(\frac{1}{2}\Big\|\hat{w} + \frac{C}{\tilde{C}}(g(\hat{w}) - \hat{w})\Big\|\right)^2.$$

The lower bound $LB(w_C^{*\top}x_i'|\hat{w}_{\tilde{C}})$ is given by solving the following optimization problem:

$$\min_{w_C^*} \quad w_C^{*\top}x_i' \quad \text{s.t.} \quad \left\|w_C^* - \frac{1}{2}\Big(\hat{w} - \frac{C}{\tilde{C}}(g(\hat{w}) - \hat{w})\Big)\right\|^2 \leq \left(\frac{1}{2}\Big\|\hat{w} + \frac{C}{\tilde{C}}(g(\hat{w}) - \hat{w})\Big\|\right)^2. \tag{18}$$

Using Proposition 8, the solution of (18) is given as

$$LB(w_C^{*\top}x_i'|\hat{w}_{\tilde{C}}) = \frac{1}{2}x_i'^\top\Big(\hat{w} - \frac{C}{\tilde{C}}(g(\hat{w}) - \hat{w})\Big) - \|x_i'\|\Big\|\frac{1}{2}\Big(\hat{w} + \frac{C}{\tilde{C}}(g(\hat{w}) - \hat{w})\Big)\Big\|$$

$$\leq \frac{1}{2}x_i'^\top\Big(\hat{w} - \frac{C}{\tilde{C}}(g(\hat{w}) - \hat{w})\Big) - \frac{1}{2}\|x_i'\|\Big(\Big|1 - \frac{C}{\tilde{C}}\Big|\|\hat{w}\| + \frac{C}{\tilde{C}}\|g(\hat{w})\|\Big)$$

$$= \begin{cases} \alpha(\hat{w}_{\tilde{C}}, x_i') - \frac{1}{\tilde{C}}(\beta(\hat{w}_{\tilde{C}}, x_i') + \gamma(g(\hat{w}_{\tilde{C}}), x_i'))C, & \text{if } C \geq \tilde{C}, \\ -\beta(\hat{w}_{\tilde{C}}, x_i') + \frac{1}{\tilde{C}}(\alpha(\hat{w}_{\tilde{C}}, x_i') + \delta(g(\hat{w}_{\tilde{C}}), x_i'))C, & \text{if } C < \tilde{C}. \end{cases}$$

Similarly, the upper bound $UB(w_C^{*\top}x_i'|\hat{w}_{\tilde{C}})$ is given by solving the following optimization problem

$$\max_{w_C^*} \quad w_C^{*\top}x_i' \quad \text{s.t.} \quad \left\|w_C^* - \frac{1}{2}\Big(\hat{w} - \frac{C}{\tilde{C}}(g(\hat{w}) - \hat{w})\Big)\right\|^2 \leq \left(\frac{1}{2}\Big\|\hat{w} + \frac{C}{\tilde{C}}(g(\hat{w}) - \hat{w})\Big\|\right)^2, \tag{19}$$

and the solution of (19) is given as

$$UB(w_C^{*\top}x_i'|\hat{w}_{\tilde{C}}) = \frac{1}{2}x_i'^\top\Big(\hat{w} - \frac{C}{\tilde{C}}(g(\hat{w}) - \hat{w})\Big) + \|x_i'\|\Big\|\frac{1}{2}\Big(\hat{w} + \frac{C}{\tilde{C}}(g(\hat{w}) - \hat{w})\Big)\Big\|$$

$$\geq \frac{1}{2}x_i'^\top\Big(\hat{w} - \frac{C}{\tilde{C}}(g(\hat{w}) - \hat{w})\Big) + \frac{1}{2}\|x_i'\|\Big(\Big|1 - \frac{C}{\tilde{C}}\Big|\|\hat{w}\| + \frac{C}{\tilde{C}}\|g(\hat{w})\|\Big)$$

$$= \begin{cases} -\beta(\hat{w}_{\tilde{C}}, x_i') + \frac{1}{\tilde{C}}(\alpha(\hat{w}_{\tilde{C}}, x_i') + \delta(g(\hat{w}_{\tilde{C}}), x_i'))C, & \text{if } C \geq \tilde{C}, \\ \alpha(\hat{w}_{\tilde{C}}, x_i') - \frac{1}{\tilde{C}}(\beta(\hat{w}_{\tilde{C}}, x_i') + \gamma(g(\hat{w}_{\tilde{C}}), x_i'))C, & \text{if } C < \tilde{C}. \end{cases}$$

∎

**Remark 9** (Extended version of Remark 5 in the main text)**.** *We note that the idea of using Propositions 7 and 8 for proving Lemma 1 is inspired from recent studies on safe screening [12, 13, 14, 15, 16]. Safe screening has been introduced in the context of sparse modeling. It allows us to identify sparse features or instances before actually solving the optimization problem. A key technique used in those studies is to bound Lagrange multipliers at the optimal solution (Lagrange multiplier values at the optimal solution tell us which features or instances are active or non-active) in somewhat similar way as we did in §3. Our main contribution is to borrow this idea for representing a validation error lower bound as a function of the regularization parameter, and show that it can be used for finding an approximately optimal regularization parameter with theoretical guarantee.*

Figure 4: An illustrative example of Algorithm 2 behavior. The blue real lines represent the validation error lower bound. The red chained lines and green dashed lines indicate the current best validation error upper bound $E_v^{\text{best}}$ and $E_v^{\text{best}} - \varepsilon$, respectively. If the blue validation error lower bound falls below the green ones, the validation error can be smaller by $\varepsilon$ than the current best. In such a case, the algorithm computes the next approximate solution, and update the validation error lower bound based on the new approximate solution. The plot is an enlarged view of the region from $\tilde{C}_{13}$ to $\tilde{C}_{17}$ in Figure 3 (a) in §5.

## B  Details of the speed-up tricks for finding an $\varepsilon$-approximate regularization parameter

In this appendix, we first describe two modifications of the basic algorithm for finding an $\varepsilon$-approximate regularization parameter presented in §4.2 for further speed-up.

**Trick1**   The efficiency of the algorithm depends on how far one can step forward in each iteration. We see in (10) that the step size $\tilde{C}_{t+1} - \tilde{C}_t$ is large if the current minimum validation error upper bound $E_v^{\text{best}}$ is small. In other words, the step size will be small until we have sufficiently small $E_v^{\text{best}}$. It suggests that, if we can find small enough $E_v^{\text{best}}$ at an *earlier* stage of the algorithm, we can reduce the total computational cost of the algorithm. In order to find sufficiently small $E_v^{\text{best}}$ as early as possible, we propose a simple heuristic approach, where we first roughly search over the entire range by a rough grid search.

**Trick2**   Our next modification for speed-up is to use

$$LB(E_v(w_C^*)|\hat{w}_{\tilde{C}_t}, \hat{w}_{\tilde{C}_{t+1}}) := \max\{LB(E_v(w_C^*)|\hat{w}_{\tilde{C}_t}), LB(E_v(w_C^*)|\hat{w}_{\tilde{C}_{t+1}})\},$$

for computing the validation error lower bound in $C \in [\tilde{C}_t, \tilde{C}_{t+1}]$. It provides a tighter validation error lower bounds than using $LB(E_v(w_C^*)|\hat{w}_{\tilde{C}_t})$ alone, meaning that larger step might be allowed in each iteration. However, we cannot actually compute $LB(E_v(w_C^*)|\hat{w}_{\tilde{C}_{t+1}})$ before we fix $\tilde{C}_{t+1}$. We thus propose a simple trial-and-error approach. Specifically, we step forward a little bit further than (10) when we select the next $\tilde{C}_{t+1}$. After we fix $\tilde{C}_{t+1}$, we compute an approximate solution $\hat{w}_{\tilde{C}_{t+1}}$ and then check whether the validation error $E_v(w_C^*)$ is not smaller by $\varepsilon$ than the current minimum for $C \in [\tilde{C}_t, \tilde{C}_{t+1}]$ by using now available $LB(E_v(w_C^*)|\hat{w}_{\tilde{C}_t}, \hat{w}_{\tilde{C}_{t+1}})$.

Algorithm 3 is the pseudo-code of the proposed algorithm along with tricks 1 and 2.

There are two additional input parameters $m \in \mathbb{N}$ and $\rho > 1$. The former is used for trick1, where we initially compute $m$ approximate solutions for regularization parameter values evenly allocated in the interval $[C_l, C_u]$ in the logarithmic scale. Trick1 is described at lines 2-9 in Algorithm 3.

The latter $\rho > 1$ is used for trick2, where the next regularization parameter value is determined in trial-and-error manner. To formally describe trick2, let us define a set $\Gamma$ as a function of $w$ in the following way

$$\Gamma(w_{\tilde{C}}) := \Big\{ \frac{\beta(w_{\tilde{C}}, x_i')}{\alpha(w_{\tilde{C}}, x_i') + \delta(g(w_{\tilde{C}}), x_i')} \tilde{C} \Big\}_{i \in \mathcal{P}} \cup \Big\{ \frac{\alpha(w_{\tilde{C}}, x_i')}{\beta(w_{\tilde{C}}, x_i') + \gamma(g(w_{\tilde{C}}), x_i')} \tilde{C} \Big\}_{i \in \mathcal{N}}.$$

Then, our initial trial step is written as

$$C^{\text{tmp}} := (\lfloor n'(LB(E_v(w_{\tilde{C}_t}^*)|\hat{w}_{\tilde{C}_t}) - E_v^{\text{best}} + \rho\varepsilon) \rfloor + 1)^{\text{th}} (\Gamma(\hat{w}_{\tilde{C}_t})), \tag{20}$$

where $\rho > 1$ represents how far we step forward. We then compute an approximate solution $\hat{w}_{C^{\text{tmp}}}$, and obtain a validation error lower bound $LB(E_v(w_C^*)|\hat{w}_{\tilde{C}_t}, \hat{w}_{\tilde{C}^{\text{tmp}}})$ by combining

---

Algorithm 3 : Finding an $\varepsilon$-approximate regularization parameter with approximate solutions using tricks 1 and 2

---

**Input:** $\{(x_i, y_i)\}_{i \in [n]}, \{(x'_i, y'_i)\}_{i \in [n']}, C_l, C_u, \varepsilon, m, \rho$
1: $C^{\text{best}} \leftarrow C_l, E_v^{\text{best}} \leftarrow 1$
2: $s \leftarrow \frac{\log_{10}(C_u) - \log_{10}(C_l)}{m}$
3: **for** $h = 0$ to $m - 1$ **do**
4:     $\bar{C}_h \leftarrow 10^{(\log_{10}(C_l) + h \times s)}$
5:     $\hat{w}_{\bar{C}_h} \leftarrow$ solve (1) approximately for $C = \bar{C}_h$
6:     **if** $UB(E_v(w^*_{\bar{C}_h}) | \hat{w}_{\bar{C}_h}) < E_v^{\text{best}}$ **then**
7:         $E_v^{\text{best}} \leftarrow UB(E_v(w^*_{\bar{C}_h}) | \hat{w}_{\bar{C}_h}), C^{\text{best}} \leftarrow \bar{C}_h$
8:     **end if**
9: **end for**
10: $\bar{C}_m \leftarrow C_u , t \leftarrow 1$
11: **for** $h = 0$ to $m - 1$ **do**
12:     $\tilde{C}_t \leftarrow \bar{C}_h , \hat{w}_{\tilde{C}_t} \leftarrow \hat{w}_{\bar{C}_h}$
13:     **while** $\tilde{C}_t \leq \bar{C}_{h+1}$ **do**
14:         Set $C^{\text{tmp}}$ by (20) using $\hat{w}_{\tilde{C}_t}$
15:         **if** $C^{\text{tmp}} > \bar{C}_{h+1}$ **then**
16:             Set $C^{\text{tmp}}$ by (22) using $\hat{w}_{\tilde{C}_t}$
17:             **if** $C^{\text{tmp}} > \bar{C}_{h+1}$ **then**
18:                 break while loop
19:             **end if**
20:         **end if**
21:         $\hat{w}_{C^{\text{tmp}}} \leftarrow$ solve (1) approximately for $C = C^{\text{tmp}}$
22:         Compute $UB(E_v(w^*_{C^{\text{tmp}}}) | \hat{w}_{C^{\text{tmp}}})$ by (8b).
23:         **if** $UB(E_v(w^*_{C^{\text{tmp}}}) | \hat{w}_{C^{\text{tmp}}}) < E_v^{\text{best}}$ **then**
24:             $E_v^{\text{best}} \leftarrow UB(E_v(w^*_{C^{\text{tmp}}}) | \hat{w}_{C^{\text{tmp}}})$
25:             $C^{\text{best}} \leftarrow C^{\text{tmp}}$
26:         **end if**
27:         $r \leftarrow 0$
28:         RecursiveCheck$(\tilde{C}_t, C^{\text{tmp}}, \hat{w}_{\tilde{C}_t}, \hat{w}_{C^{\text{tmp}}}, r)$
29:         $\tilde{C}_{t+r+1} \leftarrow C^{\text{tmp}}, \hat{w}_{\tilde{C}_{t+r+1}} \leftarrow \hat{w}_{C^{\text{tmp}}}$
30:         $t \leftarrow t + r + 1$
31:     **end while**
32: **end for**
**Output:** $C^{\text{best}} \in \mathcal{C}(\varepsilon)$.

---

$LB(E_v(w^*_C) | \hat{w}_{\tilde{C}_t})$ and $LB(E_v(w^*_C) | \hat{w}_{C^{\text{tmp}}})$. For accepting this trial step, we need to make sure that the lower bounds are not smaller by $\varepsilon$ than the current best $E_v^{\text{best}}$ for any $C \in [C_t, C^{\text{tmp}}]$. To this end, we investigate where the two lower bounds $LB(E_v(w^*_C) | \hat{w}_{\tilde{C}_t})$ and $LB(E_v(w^*_C) | \hat{w}_{C^{\text{tmp}}})$ go below $E_v^{\text{best}} - \varepsilon$. To formulate this, let us define the following two functions

$$C^R(\hat{w}_{C(L)}) := (\lfloor n'(LB(E_v(w^*_{C(L)}) | \hat{w}_{C(L)}) - E_v^{\text{best}} + \varepsilon) \rfloor + 1)^{\text{th}}(\Gamma(\hat{w}_{C(L)})), \quad (21)$$

$$C^L(\hat{w}_{C(R)}) := (\lfloor n'(LB(E_v(w^*_{C(R)}) | \hat{w}_{C(R)}) - E_v^{\text{best}} + \varepsilon) \rfloor + 1)^{\text{TH}}(\Delta(\hat{w}_{C(R)})), \quad (22)$$

where, for the latter, we define

$$\Delta(w_{\tilde{C}}) := \left\{ \frac{\alpha(w_{\tilde{C}}, x'_i)}{\beta(w_{\tilde{C}}, x'_i) + \gamma(g(w_{\tilde{C}}), x'_i)} \tilde{C} \right\}_{i \in \mathcal{P}} \cup \left\{ \frac{\beta(w_{\tilde{C}}, x'_i)}{\alpha(w_{\tilde{C}}, x'_i) + \delta(g(w_{\tilde{C}}), x'_i)} \tilde{C} \right\}_{i \in \mathcal{N}},$$

and denote the $k^{\text{TH}}$-largest element of $\Delta$ as $k^{\text{TH}}(\Delta)$ for any natural number $k$. The trial step to $C^{\text{tmp}}$ is accepted if

$$C^L(\hat{w}_{C^{\text{tmp}}}) < C^R(\hat{w}_{\tilde{C}_t}).$$

---
Algorithm 4 : RecursiveCheck $(C(L), C(R), \hat{w}_{C(L)}, \hat{w}_{C(R)}, r)$
---

Compute $C^R(\hat{w}_{C(L)})$ in (21).
Compute $C^L(\hat{w}_{C(R)})$ in (22).
**if** $C^L(\hat{w}_{C(R)}) < C^R(\hat{w}_{C(L)})$ **then**
    return
**else**
    $r \leftarrow r + 1$
    $\tilde{C}_{t+r} \leftarrow \frac{1}{2}(C^L(\hat{w}_{C(R)}) + C^R(\hat{w}_{C(L)}))$
    $\hat{w}_{\tilde{C}_{t+r}} \leftarrow$ solve (1) approximately for $C = \tilde{C}_{t+r}$
    **if** $UB(E_v(w^*_{\tilde{C}_{t+r}})|\hat{w}_{\tilde{C}_{t+r}}) < E_v^{\text{best}}$ **then**
        $E_v^{\text{best}} \leftarrow UB(E_v(w^*_{\tilde{C}_{t+r}})|\hat{w}_{\tilde{C}_{t+r}})$
        $C^{\text{best}} \leftarrow \tilde{C}_{t+r}$
    **end if**
    RecursiveCheck$(C(L), \tilde{C}_{t+r}, \hat{w}_{C(L)}, \hat{w}_{\tilde{C}_{t+r}}, r)$
    RecursiveCheck$(\tilde{C}_{t+r}, C(R), \hat{w}_{\tilde{C}_{t+r}}, \hat{w}_{C(R)}, r)$
**end if**

---

If not, we need to shrink the trial step by using the procedure described in Algorithm 4. Briefly speaking, Algorithm 4 conducts a bisection search until we find two approximate solutions $\hat{w}_{C(L)}$ and $\hat{w}_{C(R)}$ that satisfy $C^L(\hat{w}_{C(L)}) < C^R(\hat{w}_{C(L)})$. We note that, with the use of trick2, the sequence of the regularization parameter values $\tilde{C}_1, \dots, \tilde{C}_T$ is not necessarily in increasing order because they are computed in trial-and-error manner.

## C  Adaptation to cross-validation setup

All the methods presented above can be straightforwardly adapted to a cross-validation (CV) setup. Consider $k$-fold CV where $n$ instances are divided into $k$ disjoint subsets $\{F_\kappa\}_{\kappa \in [k]}$ with almost equal size. Let $w(\kappa)_C^*$ be the optimal solution trained without using the instances in $F_\kappa$. Then, the $k$-fold CV error is defined as

$$E_{k\text{CV}}(C) := \frac{1}{n} \sum_{\kappa \in [k]} \sum_{i \in F_\kappa} I\big(y_i w(\kappa)_C^{*\top} x_i < 0\big),$$

where, note that, the CV error is not a function of $w$, but a function of $C$. Our algorithm can find an $\varepsilon$-approximate regularization parameter at which the $k$-fold CV error is guaranteed to be no greater by $\varepsilon$ than the smallest possible $k$-fold CV error. For each of the $k$ folds, we can compute a validation error lower bound as described before. A lower bound of the entire $k$-fold CV error can be obtained by simply summing them up.