[Reviews · NeurIPS 2015]

Submitted by Assigned_Reviewer_1

For the important goal of test error guarantees, the combination of ideas from screening together with approximate regularization paths as proposed here is completely novel, and nicely surprising. While the result here is only for binary linear classification, it might be a useful first step towards more general ML models. The paper is clearly written, and the experimental results are encouraging.

For the reader it might be nice to more clearly distinguish explicit definitions of eps-approximate solutions in terms of training error (1) and validation error (2), to avoid confusion. (and also 'eps-approximate regularization parameter'. maybe better call it 'eps-approximate best regularization parameter'?)

The authors should discuss the theoretical complexity of the proposed approach as a function of eps more in detail. More concretely in this sense, assume the C values would be chosen naively and equally-spaced, what do your results say about the interval size? This would also be a good contact point to discuss the connection to the path methods for training error (vs text as done here) more in detail.

minor: l061: the -> this l264 soluitons

(this is a light review)
Summary: The paper contributes a novel approach to compute guarantees on the cross-validation error along the entire regularization path. This is a highly relevant first step towards much stronger methods for hyperparameter search, such that those methods could in the future come with guarantees for test error.

Submitted by Assigned_Reviewer_2

The authors propose a novel framework for computing a lower bound of the CV errors as a function of the regularization parameter. The main motivating question of the manuscript ("it is hard to tell how many grid-points would be needed in cross-validation") has been already extensively studied both from theoretical and empirical perspective. For example, see (Rifkin et al, MIT-CSAIL-TR-2007-025) or (Rasmussen and Williams, Gaussian Processes for Machine Learning, Chapter 5.)

The proposed score and validation error bounds are technically sound. These results are direct application of the ideas from safe screening (bounding Lagrangian multipliers at the optimal solution). While this was clearly stated in the supplemental material a similar remark in the main body (e.g. section 3) of the manuscript could be helpful for the reader.

The experimental setup to evaluate a theoretical approximation of the regularization parameter is not clearly presented. How exactly the grid search is performed: 10 fold cross-validation on the training set to obtain optimal parameters with final evaluation on the validation set? In that case, I assume that E_{v} on the training set should have been reported in Fig(3) instead of validation.

Summary: The proposed score and validation error bounds are technically sound. Novelty is incremental and experimental setup/result is dissapointing.

Submitted by Assigned_Reviewer_3

Interesting paper, with a potentially useful result.

It is not clear that the bound would remain valid if the method were kernelized and the kernel parameters also tuned using cross-validation (as the kernel and regularization parameters tend not to be independent).

The experimental evaluation is a little weak as there are only a few of small datasets.

The size of the datasets is especially an issue as the paper discusses the computational practicality of the bound.

It is not clear that the bound enables better models to be selected, than would be selected simply by optimizing the cross-validation error directly (within some computational budget).

The method may be useful in avoiding over-tuning the regularization parameter (i.e. over-fitting the cross-validation estimate).
Summary: The paper presents a bound on the cross-validation error for linear models and a procedure for finding a value with guaranteed CV error.

The paper is interesting, and well written, but the experimental evaluation is a little weak.

Submitted by Assigned_Reviewer_4

A common task in machine learning is to learn the regularization parameter of a model. Usually this is done using cross-validation. The main result of the paper is that is presents a new way to analytically compute bounds on validation error. First, a lower and an upper bound for wx_i is computed for each validation instance x_i (w is the parameter of the model) as a function of the regularization parameter, given a solution for a different regularization parameter. Then, the scores for validation instances are combined to yield the bounds on the validation error. A central result in the paper is Lemma 1, that shows how compute the bounds for wx_i. The proof is not given in the paper, but the reader is referred to the Appendix. Giving intuition of this core result in the main text would be very important for the reader to be able to understand the idea without going to the Appendix, even if it's entirely reasonably to omit the full proof from the main text. The proof (in the appendix) seems to rely heavily on the convexity of the regularized loss function, limiting the usability (although being more general than presented before). Also, L2 prior on the parameter is used. Furthermore, the paper only deals with tuning a univariate hyper-parameter. The results appear correct, though I did not check the proofs in detail. The text is clearly written (though see the remark above). The experiments show that the regularization parameter can be optimized efficiently by assuming a reasonably tight error bound and, importantly, that the error bound is available in the first place. The topic seems relevant for the NIPS community, and the results interesting, though potentially too specific to raise wide interest.
Summary: The paper presents analytical bounds for cross-validation error and demonstrates how these can be used to optimize the parameters efficiently.

Author Feedback
Author rebuttal: Thank you for fruitful comments.

[Comment on the scope of the paper]

Let us first make a comment about the scope of our paper because multiple reviewers pointed out that the class of problems we considered here is narrow. We would like to rebut this criticism from the following two points.

a) Some reviewers might feel that L2-penalty term is too restrictive. We would like to emphasize that this restriction is identical with assuming that the loss function is strongly convex. A huge body of theoretical works on convex empirical risk minimization problems have been devoted to the problems with strongly convex loss functions. If the reviewers claim that the scope of our work is narrow, the same criticism should be applied to those past works targeted to strongly convex loss functions. In addition, note that our approach can be used with any other convex regularization terms such as L1-regularization as long as the loss function is strongly convex.

b) The class of problems we consider here contains several important classification algorithms such as SVM and logistic regression. We would also like to emphasize that all the methodologies presented in the paper can be kernelized if the loss function can be kernelized (it means that our methodologies can be applied to kernel SVM and kernel logistic regression etc.). Furthermore, it is easy to confirm that most of the methodologies presented here can be adapted to other types of problems such as regression.

We would highly appreciate if the reviewers re-consider whether they still feel the scope of our problems is narrow when the above two points are taken into account.

[Answers to Rev 1]

> The main motivating example has been already studied ...

We agree with the reviewer in that, CV error for special problems (such as LOOCV error in least-square regression) can be represented as a function of the regularization parameter, and it is possible to provide approximation guarantee of CV error in these special cases. Our contribution can be interpreted as an extension of these special cases to wider class of problems (namely, problems with strongly convex loss functions). We believe that this extension is non-trivial. We will discuss and clarify this point in the final version.

> The experimental setup is not clearly presented:

Fig 2 is devoted for illustrating Problem 1 in sec 4.1. In Fig 2, we compared three grid search methods: 1) simple grid search at evenly allocated regularization parameters in logarithmic scale (red), Bayesian optimization (blue), and Algorithm 2 (green). On the other hand, Fig 3 and Tab 1 are devoted for illustrating Problem 2 in sec 4.2. In Fig 3 and Tab 1, we used Algorithm 2. Since Fig 3 is only for illustration, we just plotted the error on independent validation set (not CV error). We apologize for the confusion, and make it clear in the final version.

[Answers to Rev 2]

> practically utility seems limited ...

I am afraid there is a misunderstanding here. Algorithm 2 automatically selects a sequence of regularization parameter values, and it is guaranteed that approximate solutions trained at these regularization parameters are shown to satisfy the desired approximation quality. For example, it could find eps=0.01 approximation-guaranteed solution for D10 data set (n > 64000) in 36.81 seconds (see Tab 1). We believe we can call it practical.

> why green-curve is good in Fig2?

Green-curve (Algorithm 2) is better since it cleverly selects the sequence of regularization parameter values. In other words, green-curve is better because it can skip ranges where solutions are shown to be no better than the curren best by \eps.

[Answers to Rev 3]

> Intuition of the proof of Lemma 1 should be provided in main text.

Thank you for the suggestion. We would do that in the final version.

> Convexity and L2-penalty seems to be restrictive assumption.

See "Comment on the scope of the paper" above.

[Answers to Rev 4]

> Clarification between eps-approximate solutions in terms of training error and validation error would be needed

Thank you for suggestion. We will clarify this point in the final version.

> theoretical complexity of the path

We understand this is important. We conjecture that we can derive theoretical complexity by using Lipschitz constant of CV error curve.

[Answers to Rev 5]

See "Comment on the scope of the paper" above.

[Answers to Rev 6]

> Is it valid when it is kernelized?

All the methodologies presented here can be kernelized if the loss function can be kernelized. See "Comment on the scope of the paper b)" above.

> There are only a few of small datasets...

In Tab 2, we summarized 10 data sets whose sample size ranges from 270 (D1) to 64700 (D10). We do not think it is "only a few of small datasets", although we can apply the method to larger datasets if it is necessary.